# The Sous Vide Cooking of Mediterranean Mussel (*Mytilus galloprovincialis*): Safety and Quality Assessment

**DOI:** 10.3390/foods12152900

**Published:** 2023-07-30

**Authors:** Giovanni Luca Russo, Antonio Luca Langellotti, Gabriele Buonocunto, Sharon Puleo, Rossella Di Monaco, Aniello Anastasio, Valeria Vuoso, Giorgio Smaldone, Marco Baselice, Federico Capuano, Francesca Garofalo, Paolo Masi

**Affiliations:** 1CAISIAL Centre, University of Naples Federico II, Via Università 133, 80055 Portici, Italy; giovanniluca.russo@unina.it (G.L.R.); ga.buonocunto@libero.it (G.B.); sharon.puleo@unina.it (S.P.); dimonaco@unina.it (R.D.M.); masi@unina.it (P.M.); 2Unit of Food Science and Technology, Department of Agricultural Sciences, University of Naples Federico II, 80055 Portici, Italy; 3Department of Veterinary Medicine and Animal Production (MVPA), University of Naples Federico II, 80137 Napoli, Italy; anastasi@unina.it (A.A.); valeria.vuoso@unina.it (V.V.); 4Centro di Riferimento Regionale per la Sicurezza Sanitaria del Pescato (CRiSSaP), 80143 Napoli, Italy; giorgio.smaldone@unina.it; 5ASL Caserta, Department of Prevention, Complex Unit Hygiene of Animal Origin Foods, 81100 Caserta, Italy; 6Department of Civil, Environmental, Land, Construction and Chemistry (DICATECh), Politecnico di Bari, 70126 Bari, Italy; marco.baselice@poliba.it; 7Department of Food Microbiology, Istituto Zooprofilattico Sperimentale del Mezzogiorno, 80055 Portici, Italy; federico.capuano@izsmportici.it (F.C.); francesca.garofalo@izsmportici.it (F.G.)

**Keywords:** *Mytilus galloprovincialis*, sous vide, semi-preserved food, sensory analysis, shelf life, seafood

## Abstract

This study involves an investigation of the effects of various cooking temperatures, freeze–thaw processes, and food preservatives on the quality and shelf-life of sous vide Mediterranean mussels. Cooking temperatures of 80 °C or above significantly improved the microbiological quality, with bacterial counts remaining within the acceptability range for human consumption even after 21 days of refrigerated storage. Fast freezing followed by slow thawing preserved the highest moisture content, potentially improving texture. Sensory analysis revealed that refrigerated sous vide mussels maintained a comparable taste to freshly cooked samples. Frozen samples reheated via microwaving exhibited more intense flavour than pan-reheated or fresh mussels. Food additives, including citric acid, potassium benzoate, and potassium sorbate, alone or in combination with grape seed oil, significantly reduced total volatile basic nitrogen and thiobarbituric acid-reactive substances during 28 days of storage, indicating decreased spoilage and lipid oxidation. Mussels with a combination of these additives registered a nitrogen content as low as 22 mg of N/100g after 28 days, well below the limit of acceptability (<35 mg of N/100g). Food additives also inhibited bacterial growth, with mesophilic bacteria count below 3.35 Log CFU/g after 28 days, compared with 5.37 Log CFU/g in control samples. This study provides valuable insights for developing optimal cooking and preservation methods for sous vide cooked seafood, underscoring the need for further research on optimal cooking and freeze–thaw protocols for various seafood types.

## 1. Introduction

*Mytilus galloprovincialis* are important edible bivalve mussels, widely distributed and farmed in the Mediterranean area. They represent an important economic income source in Europe. Spain is the largest mussel-producing country in Europe, with an annual production of more than 200 KT [1], and Italy is in second place with about 60 KT/y. Nevertheless, about 70.000 T of mussels are imported into Italy each year (mainly from Spain) to meet the market demand [2]. Mussels represent an important gastronomic tradition in Italy, and they are mainly consumed fresh, and a very small percentage of these are destined for processing.

Fresh mussels are highly perishable, even under refrigerated conditions. This perishability limits their shelf life and poses a challenge to meet the market demand, especially in regions where these mussels are not locally sourced. The transformation processes must take into account two essential factors: to increase the shelf life of the product and to have a product with acceptable taste and flavour. Mussel processing enables the commercialisation of products for new consumers and new markets while extending their limited shelf life [3]. However, the current production processes (i.e., freezing) of seafood products affect their overall quality, decreasing their organoleptic characteristics and most of all their texture and flavour [4]. This presents a significant gap in the industry: the need for a processing method that not only extends the shelf life of mussels but also maintains their fresh taste and flavour. Many studies have been conducted on novel methods for processing fresh mussels, such as cooking and vacuum [5], smoking [6], canning [7], frying or baking stuffed products [8], and pasteurisation in red sauce [9].

In recent years, there has been a growing international interest in the sous vide preparation of seafood [10]. The sous vide technique is one of the alternative production processes capable to fulfil consumer demands for ready-to-eat foods [11,12] and could be a potential solution for extending the shelf life of seafood products without affecting their organoleptic characteristics. This cooking technique consists of placing foods in vacuum-sealed bags and cooking them at low temperatures (60–90 °C) for a longer period than conventional methods like boiling or roasting. The main advantage of using this technique consists of an improved tenderness, colour retention, and flavour of the cooked products [13] along with a reduction in lipids oxidation, protein damage, and other heat-sensitive compounds. On the other hand, sous vide technology has raised concerns about the microbiological safety of products cooked following inadequate pasteurisation processes [14]. The microbiological quality of vacuum-packed products is still a field to be explored as there are also studies that report better microbial stability in sous vide food than in food prepared using traditional methods [15].

Recently, Bongiorno et al. (2018) evaluated the chemical, microbiological, and sensory quality during chilled storage (+3 °C) of sous vide mussels. However, the physical properties and the opportunity to freeze the vacuum-packed product were not evaluated in this study. Moreover, the sensory differences between various reheating methods have never been studied for vacuum-packed mussels before. Indeed, several studies, suggested that the sensory characteristics of precooked fish products could be affected by the type of reheating process [16,17].

The main issue concerning the development of new seafood products is the risk of pathogenic microorganisms such as hepatitis A virus, *Salmonella* spp., *Listeria monocytogenes,* and *Clostridium botulinum*. These pathogens can be effectively removed via heat treatments with a proper range of time/temperature, as stated by the European Food Safety Authority (EFSA) [18].

From this perspective, this study aims to pioneer novel methodologies for creating sous vide mussels that ensure microbiological safety and prolonged high-quality organoleptic characteristics. Specifically, the research objective is the evaluation of total volatile basic nitrogen (TVB-N), thiobarbituric acid reactive substances (TBARSs), and physical attributes, as they serve as critical indicators of product quality and freshness. Furthermore, this study involves an in-depth assessment of the microbiological quality of sous vide mussels under varying cold storage conditions. The overarching goal is to quantify the correlation between these analytical metrics and the sensory properties of mussels. By creating an integrative framework of these variables, the study seeks to formulate best practices for the sous vide preparation and storage of mussels that offer high quality and safety standards.

## 2. Materials and Methods

### 2.1. Mussel Samples and Experimental Procedure

Live refrigerated mussels (*Mytilus galloprovincialis*) farmed in Campania Region were purchased at a fish market in Naples (Agro-Food Center C.A.A.N, Volla, Italy). The mussels were transported in the laboratory in about thirty minutes using a refrigerated polystyrene box and placed in a refrigerated chamber (+4 °C) before being processed. They were manually sorted to eliminate dead or damaged molluscs and subjected to cleaning and shell washing. The mussels were then classified based on weight and size. For the experiments, only mussels with a length of more than 6 cm and an average weight of 22 ± 5 g were used.

To standardise the cooking process, meat yield and the condition index were assessed in all the batches of selected mussels. The condition index was determined according to Peharda et al. (2007) on 30 mussels of each batch and was defined as the percent ratio between cooked meat weight and the sum of cooked meat weight and shell weight. The meat yield (MY) was calculated as the percent ratio of wet meat weight to total live weight [19]. Only mussel batches with CI and MY ranging from 25% to 35% were used in the experiments.

Mussel samples of 500 ± 10 g were vacuum-sealed in 20 × 30 cm PA/PP sous vide cooking bags (MyVac, Niederwieser Spa, Campogalliano, Italy) designed for vacuum storage and sous vide cooking up to 121 °C for 60 min. The mussels were then cooked submerged in a thermostatic bath (SIRMAN Softcooker, Quarto D’Altino–(VE), Italy) at four different times/temperatures: 72 °C for 450 s, 80 °C for 270 s, 90 °C for 90 s, and 100 °C for 40 s. The cooking curves were chosen on the basis of the recommendations of the EFSA panel on biological hazards regarding the thermal treatment that could be applied to bivalve molluscs to eliminate pathogenic microorganisms such as hepatitis A [20].

The attainment of the target internal temperature was ascertained using a thermocouple probe, which was carefully inserted into the centre of a mussel through a small perforation made in the shell. To promptly halt the cooking process, the samples were swiftly removed and placed in a blast chiller set to a temperature of −1 °C for a duration of 15 min. Once cooled, the samples were stored in a refrigerated environment maintained at a constant temperature of +4 °C. Analytical samples were systematically taken at predetermined time intervals, specifically at the commencement of the storage period and subsequently on the 7th, 14th, 21st, and 28th days. The best procedure of sous vide cooking in terms of microbial and mechanical quality was also used to prepare a frozen product. Cooked mussels were subjected to three different freezing curves from +4 °C to −20 °C: 30 min (fast freezing, FF); 45 min (medium-speed freezing, MF); 60 min (slow freezing, SF). The samples for analysis were taken after 14 d from storage at −18 ± 1 °C. Two thawing methods were assessed: fast thawing (FT) in a microwave (Samsung MC35R8088LC/ET, Seoul, Korea) at 750 W for 180 s and slow thawing at +4 °C for 16 h. Reheating for both treatments was performed until reaching a mean internal mussel temperature of +65 ± 3 °C.

After the preliminary trials, to increase the shelf-life of the refrigerated product, the optimal sous vide cooking procedure was used to test different preservative treatments using citric acid, grape seed extract, potassium benzoate, and potassium sorbate. These additives were added in a liquid form from stock solutions, and aliquots were pipetted into the bags with mussels and shaken before cooking. The following formulations were used in the samples: (a) pH of up to 4.6 by adding citric acid (1 g/kg) in the vacuum bag before the cooking process (AC); (b) pH of up to 4.6 by adding citric acid and 20 g/kg of grape seed oil as antioxidant (BC); (c) pH of up to 4.6 by adding citric acid and 1 g/kg of potassium benzoate as antimicrobial agent, and 1 g/kg of potassium ascorbate as antioxidant (CM). The selected concentration of additives was chosen after preliminary trials on microbial and chemical quality over time. The quantities of the selected preservatives were all below the legal limits imposed by the European community for each compound.

TVB-N, TBARS, and microbiological analyses were performed after 0, 7, 14, 21, and 28 days of refrigerated storage on the preservative treatments and assessed against a control group (control) without any compound added but applying the same cooking and storing conditions.

### 2.2. Physical Properties

The shear force of mussel meat was determined using a texture analyser equipped with a 6 cm large blade (TMS-Pro Texture Analyser, Food Technology Corporation, Sterling, VA, USA) and a load cell of 50 N applying a 0.05 N preload. The test was conducted on 15 single-shelled mussels for each treatment. The data were calculated and analysed with Texture Lab Pro Software (version 1.1).

The colour analysis, as a quality indicator of mussels, was studied using an IRIS high-resolution imaging visual analyser (IRIS–Alpha M.O.S. Visual Analyser VA400–500x, Toulouse, France). Colour values were expressed as L*a*b* values according to the method of Jeon et al. (2020) on 15 mussels for each treatment [21].

Finally, the moisture of cooked mussels, which is relevant for the assessment of juiciness, was gravimetrically assessed by drying 10 shelled and drained mussels from each treatment in an oven at 105 °C for 16 h.

### 2.3. Chemical Analysis

TVB-N content was determined according to the method described by Sadok et al. (1996) [22] on 5 mussels from each treatment. The samples were prepared through the homogeneous grinding of mussel meat. Subsequently, 10 g of the ground sample was mixed with 90 mL of a perchloric acid solution (6 g/100 mL) and homogenised with a mixer. A steam distillation was then performed, using 50 mL of the previously obtained extract. The alkalinisation was controlled using phenolphthalein (1 g/100 mL in 95% ethanol), and then a silicone antifoaming agent and 6.5 mL of a sodium hydroxide solution (20 g/100 mL) were added. The apparatus was adjusted so that approximately 100 mL of distillate was produced in 10 min. The distillation effluent tube was immersed in a tank containing 100 mL of boric acid solution, to which 3–5 drops of the indicator solution were added. After exactly 10 min, distillation was terminated. The volatile bases contained in the solution of the accumulation tank were determined via titration with a standard solution of hydrochloric acid.

The determination of 2-thiobarbituric acid was based on the spectrophotometric quantitation of the pink complex formed after the reaction of one molecule of malondialdehyde (MDA) with two molecules of 2-thiobarbituric acid (TBA), according to the method of Goulas and Kontominas (2005). TBARS values were expressed in units of mg/malonaldehyde/kg sample.

### 2.4. Microbiological Analysis

Microbiological analyses were performed on refrigerated products 0, 7, 14, 21, and 28 days after cooking. Analyses were carried out to evaluate the presence/absence of the following pathogenic microorganisms: *L. monocytogenes* (ISO 11290-1:2017) and *Salmonella* spp. (ISO 6579-1:2017).

For all the tests, 25 g of each whole sample was diluted 1:10 in saline–peptone water sterile bags and homogenised using a stomacher.

The determination of *Enterobacteriaceae* was performed according to standard UNI-EN-ISO 21528-2:2017 [23] using Violet Red Bile Glucose Agar (VRBGA-Oxoid CM0485). *E. coli* was determined according to the ISO standard 16649-3:2015. Total mesophilic and psychotropic bacteria were determined according to the ISO standard UNI EN ISO 4833-1:2022 using Plate Count Agar (PCA, Oxoid, Milan, Italy). Coagulase-positive staphylococci were determined according to UNI ISO 6888-1:2018 using Baird Parker Agar (BPA, Oxoid, Milan, Italy). Mesophilic lactobacillus (LAB) was determined according to the ISO standard 15214:2015 using de Man, Rogosa, and Sharpe Medium (MRS, Oxoid CM0361) after incubation at 25 °C for 2 days. *Pseudomonas* was enumerated on Cetrimide Fusidin Cephaloridine Agar (CFC, Oxoid CM 559) and yeast and mould on Yeast Extract Glucose Chloramphenicol Agar (YGC–Merck, Rowey, NJ, USA, 1.16000). Microbiological data were transformed into logarithms of the number of colony-forming units (CFU/g). All plates were examined visually for typical colony types and morphology characteristics associated with each growth medium. The analyses were performed in triplicate agar plates on serial decimal dilutions of each mussel homogenate.

### 2.5. Sensory Analysis

The sensory analyses were performed according to the method proposed by Turan et al., (2007) [24], with minor modifications. In total, 15 assessors (8 males and 7 females; mean age = 34 ± 6 y.o.) were selected for their sensory acuity and trained (5 h of training sessions) to evaluate the sensory attributes of the product. The descriptors selected were brightness, typical mussel odour, fresh odour, turgidity, gumminess, succulence, sweet taste, salty taste, and typical mussel taste. Table 1 reports the list of attributes and the references used during the training for the correct use of the scale. Each characteristic was scored using a point scale (0–10), where 0 is “absent”, 10 is “very intense”, and 5 is “moderate”.

Two different sessions were performed to evaluate the chilled stored sous vide mussels and frozen mussels. The chilled stored product was evaluated after two reheating treatments: in a pan with a lid for 2 min at medium-high flame and by placing it directly in a vacuum bag in a microwave at 650 W for 1 min. The frozen product was evaluated for two reheating processes: in a pan with a lid for 3 min at medium-high flame and by placing it directly in a vacuum bag in a microwave at 650 W for 2 min. These reheating methods were chosen as they are the most common homemade methods for these kinds of products [17]. Both reheating methods allowed the temperature at the core of the mussel to reach about 90 °C, which was measured in preliminary trials using a thermal probe.

In both panel trials, a comparison with live fresh mussels was also performed in order to observe the sensory differences between sous vide mussels and fresh products. The fresh mussels were also cooked regularly on a pan with a lid until all the shells were open.

### 2.6. Statistical Analysis

All treatments were carried out at least in triplicate, and for the analyses, average values with standard deviations were reported. One-way ANOVA was carried out using raw data to test for significant differences among the samples (the significance level was always set at *p* < 0.05). Tukey’s test was used for post hoc analysis when there were significant differences among the samples. In cases in which samples were compared in pairs, a two-sample *t*-test was used. Data were analysed using IBM© SPSS© Statistics software Ver. 23 (SPSS, Inc., Chicago, IL, USA).

## 3. Results and Discussion

### 3.1. Physical Properties of Sous Vide Cooked Mussels

The results of the physical properties of sous vide cooked mussels are reported in Table 2. Significant differences were observed in terms of moisture. In particular, by comparing the fresh samples (0 d), a decreasing trend of moisture was observed as a function of the increasing treatment temperature. These results were quite expected since with an increase in temperature, the water loss tends to be faster and more intense [25]. By contrast, when comparing the samples after 21 days of storage (21 d), a long-term effect on the moisture was observed. In particular, with the increase in the treatment temperature, the moisture tended to be more preserved.

In terms of lightness (L*), yellowness (b*), and redness (a*), some significant differences were observed between the samples cooked at various treatment temperatures (*p* < 0.05). The lightness parameters remained stable over time, except for the samples treated at 90 and 100 °C in which there was a marked decrease in this parameter. For the red/green value instead (a*), the samples cooked at 72 °C showed significant differences from those cooked at 90 and 100 °C. From a colorimetric point of view, the samples cooked at 72 and 80 °C were those that exhibited greater stability than the other samples with higher temperatures. Colour changes in shellfish are often a result of mixed complex reactions such as protein denaturation, lipid oxidation, and Maillard reaction [26]. An increase in the b* value (indicating a shift towards yellow) has been associated with lipid oxidation in seafood. This is because lipid oxidation can lead to the formation of compounds that absorb light in the blue region of the spectrum, causing the food product to appear more yellow [27]. In our case, a higher temperature led to an increase in the b* values. Moreover, an increase in the treatment temperature had a negative effect on the brightness of the product due to protein denaturation. In fact, at the beginning of the storage period, we observed a significant increase in the L values (Table 2) when the treatment temperature increased (87.4 at 72 °C vs. 93.6 at 100 °C).

Figure 1 shows the trend of the shear force values measured on fresh and 21-day-stored mussels treated at different temperatures.

Firstly, the shear force values extrapolated after 21 days of refrigerated storage were significantly lower than those of the fresh products. Secondly, the temperature treatment did not affect the shear force at the beginning of the storage period (0 d), but it significantly affected this parameter after 21 days of storage. This result is in accordance with the moisture trends shown in Table 2. With the increase in moisture, the shear force tended to be lower [28].

The term “shear force” in food technology is defined as any cutting action that splits a product into two fragments. In the case of meat and fish, the most important technological parameter is the one that influences the structure and conformation of myofibrillar and connective tissue proteins, i.e., the temperature of cooking [29]. In fact, many authors stated that an alteration in myofibrillar and connective tissue proteins is associated with the heating of fish muscle after rigor mortis [29,30]. It is also reported that the shear force increases proportionally with temperature up to a maximum of 70 °C, after which a plateau occurs. This could explain why, in our experiment, the shear force between the samples cooked at 72 and those cooked at 100 °C was not significantly different at the beginning of the storage.

The effects of different combinations of freezing and thawing on the physical properties of sous vide cooked mussels are reported in Table 3. The mussels for freezing trials were cooked at 80 °C for 270 s.

The L value represents lightness, with higher values indicating lighter colours. Fast freezing followed by slow thawing (FF-ST) and medium freezing followed by slow thawing (MF-ST) resulted in the lightest colour (72.9 and 73.0, respectively), while slow freezing followed by slow thawing (SF-ST) resulted in the darkest colour (67.0). The a value represents the green-red colour spectrum. More negative values indicate a shift towards green, while more positive values indicate a shift towards red. The mussels subjected to slow freezing, followed by either fast or slow thawing (SF-FT and SF-ST), showed a stronger shift towards green (−6.4 and −6.3, respectively) than the mussels subjected to medium freezing followed by slow thawing (MF-ST, −4.6). The b value represents the blue-yellow colour spectrum. More positive values indicate a shift towards yellow. Medium freezing followed by fast thawing (MF-FT) resulted in the most yellow colour (41.7), while slow freezing followed by slow thawing (SF-ST) resulted in the least yellow colour (37.5). As reported by Sun et al., 2021, slow thawing also worsened the b* value of carp, which was mainly caused in response to the loss of carp juice and oxidation of the pigmented proteins [31]. In fact, it is widely reported that lipid oxidation and pigment degradation during thawing can lead to changes in meat colour [31,32].

The moisture content was highest in mussels subjected to fast freezing followed by slow thawing (FF-ST, 71.9%) and lowest in mussels subjected to slow freezing followed by slow thawing (SF-ST, 67.4%). Moisture content varied significantly, with the highest in mussels undergoing fast freezing followed by slow thawing (FF-ST), suggesting that this combination might better preserve the water content in mussels, potentially leading to a juicier texture.

Shear force was highest in mussels subjected to fast freezing followed by slow thawing (FF-ST, 6.1 N) and medium freezing followed by slow thawing (MF-ST, 6.0 N). The lowest shear force was observed in mussels subjected to medium freezing followed by fast thawing (MF-FT, 4.0 N).

It is reported that the formation of ice crystals physically destroys the cell structure during the frozen storage of mussels, resulting in the release of cytoplasm after thawing. Consequently, frozen storage decreases the product texture [33]. Moreover, many studies reported that freezing and thawing processes can significantly affect the texture of seafood. For instance, a study on Pacific white shrimp revealed that repeated freeze–thaw cycles could degrade the texture of meat, mainly due to protein oxidation and denaturation, as well as moisture redistribution. These effects were especially pronounced after three freeze–thaw cycles [34].

With these assumptions, the mussels treated with FF-FT were selected for subsequent sensorial analysis in order to test the mussels with less damage due to freezing.

### 3.2. Assessing Mussel Quality: TVB-N and TBARS Measurements

TVB-N is widely used to determine the freshness of seafood products in the industry. TVB-N content of sous vide cooked mussels is reported in Figure 2.

The initial TVB-N content of samples is an indicative value of freshness for raw fish materials [35]. However, several studies have underlined the poor correlation of TVB-N with the time of storage and with the sensory quality of fish [36,37].

In our case, at the initial storage time, the TVB-N value ranged between 6 and 8 mg N/100 g, which is in agreement with [38], who reported an initial TVBN content in raw mussels at 15 mg N/100 g and 9–10 for cooked mussels at the initial storage time. As shown in the figure, different time/temperature treatments of vacuum-packed products showed similar trends, without significant differences between the treatments (*p* > 0.05). For all treatments, there was a gradual increase in TVB-N from 0 to 14 days, reaching values higher than 30 mg N/100 g after 21 days of storage. This is in contrast to the work of Bongiorno et al., which stated that after 50 days of refrigerated storage, the TVB-N values of vacuum-packed mussels remained constant. However, the authors added brine solutions and stored the mussels at +3 °C, which could affect the TVB-N content during storage. In our case, the mussels were packed without any brine solution or preservatives. Nevertheless, our data are in agreement with [39], which reported instead a TVB-N value of 30 mg N/100 gr for mussels vacuum-packed after 14 days of chilled storage. In our study, the TVB-N content in mussels exceeded the limits of acceptability imposed by the European Community Directive 95/149 (35 mg N/100 g) only after 21 days of storage. The TVB-N values of seafood products increase accordingly with the growth of microorganisms; in fact, TVB-N mainly results from the metabolic activity of spoilage microorganisms [40].

The results of the TBARS analysis are reported in Figure 2. As for TVB-N, the TBARS values also increased gradually over time, indicating the occurrence of lipid oxidation. The mussels cooked at 72 °C were reported to have the lowest initial values of TBARS (0.133 mg MDA/kg) and the highest after 21 days of storage at 4 °C (0.147 mg MDA/kg). Conversely, the mussels cooked at 80 °C were those that recorded the lowest TBARS value after 21 days of refrigerated storage at +4 °C. However, we did not observe any significant difference between the samples cooked at 80 °C and 90 °C. On the other hand, the mussels cooked at 100 °C were the ones that showed the highest initial value of TBARS (0.202 mg MDA/kg), with a high content also after 21 days. These data are also in line with the colorimetric analysis (Table 2), in which the highest b* value was observed with the mussels cooked at 100 °C, denoting an increase in lipid oxidation. A TBARS value between 0 and 2 is usually associated with not rancid fish meat, while values higher than 2 mg MDA/kg indicate a rancid product [41,42]. In our study, only after 21 days, the TBARS values exceeded 1 mg MDA/kg. However, none of the samples exceeded the maximum acceptability value.

The increase in TBARS values in mussel meat during cold storage indicates the decomposition of hydroperoxides into the secondary oxidation products during lipid oxidation [43]. The lipid degradation in shellfish usually is attributed to the action of lipases present inside mussel muscles (enzyme-catalysed oxidation) but also to the autoxidation of PUFA, which undergoes a spontaneous oxidation process without light and catalyst [44]. This explains why a significant increase in TBARS values was observed even in vacuum-packed mussels, as already observed by Goulas et al. (2008) [39]. In fact, the authors stated that the type of microbiota present in the mussels, and the CO_2_ dissolved in the tissue, could also favour the self-oxidation of PUFA.

### 3.3. Microbiological Quality of Refrigerated Sous Vide Mussels

The changes in microbiological quality during the storage period are reported in Table 4.

The highest total mesophilic bacteria (TMB) content was observed for samples cooked at 72 °C, with a total bacteria content of 1.21 Log CFU/g after 7 days of refrigerated storage (+4 °C), and reaching a value of 5.73 after 21 days. At 80, 90, and 100 °C cooking temperatures, mesophilic bacteria were detectable after 14 d of storage, with an exponential increase up to 21 days.

The ICMSF (1986) recommends the value of 7 Log CFU/g as the upper acceptability limit of the total viable count (TVC) for freshwater fish and seafood [45]. For cooked foods, the acceptable limit for human consumption is reached if the total bacterial count is below 5 Log CFU/g [46]. In our study, only the mussels cooked at 72 °C were above this limit after 21 d of storage. In general, the samples cooked at 72 °C showed the worst microbiological quality, as they also suggested the presence of *Pseudomonas* sp. and LAB, which were absent in all other samples. The LAB count was determined because LAB has been reported to be the dominant spoilage flora in vacuum-packaged mussels [47]. In our case, the presence of LAB was noted only in the samples cooked at 72 °C, reaching a microbial count of 2.13 log CFU/g after 21 days. This result is also in line with other studies on mussels [38].

From a microbiological perspective, our study revealed that the samples cooked at temperatures higher than 80 °C did not show notable differences, thus suggesting that cooking at 80 °C was sufficient to stabilise the product. Our results are in line with those from Bongiorno et al. (2018), in particular for the spoilage microorganism count. In fact, in our study, *Pseudomonas* sp., yeast, and moulds were inhibited when the samples were cooked sous vide at a temperature higher than 80 °C. This was also in agreement with the results of Rhodehamel (1992), which stated that *Pseudomonas* growth and yeast growth are prevented when cooking under sous vide conditions [48].

Additionally, in our study, we ascertained that potentially harmful pathogens such as anaerobic sulphite-reducing clostridia, *Listeria monocytogenes*, and *Salmonella* were absent in all the mussel samples analysed.

### 3.4. Results of Sensorial Analysis

Figure 3 illustrates the results of the sensorial analysis performed on the refrigerated (after 14 d of storage at +4 °C) and frozen (after 14 days of storage at −18 °C) sous vide samples.

An important contribution of our study is the sensory analysis of sous vide mussels under varying cold storage conditions. The cooked and refrigerated samples did not show any significant difference from the fresh mussels cooked using the traditional method (control). Under these conditions, all the samples reported a high succulence, with a low level of graininess. These results are in line with those of Bongiorno et al. (2018), which reported no significant differences between the sous vide cook–chill mussels and conventionally cooked mussels after 14 days of storage (3 °C) [38]. In the above-mentioned work, the authors compared sous vide cook–chill mussels and mussels cooked with the conventional method and then placed in OPA/PP pouches at different storage times. In our case, we compared fresh mussels cooked immediately before the sensorial analysis and sous vide cook–chill mussels stored for 14 days at 4 °C and −20 °C. This approach allowed us to compare the difference between the fresh and the preserved mussels more realistically. Considering the frozen mussels, a significant effect was observed on the typical mussel taste. In particular, the samples reheated in the microwave were perceived with a more intense typical mussel taste than pan-reheated samples and the control. It is well known that freezing well preserves the flavours of food. Moreover, different studies showed that reheating in a microwave allows for the better preservation of the sensorial aspects, especially in terms of flavour, compared with other reheating methods [16,17]. In fact, we found that refrigerated sous vide mussels maintained a taste comparable to freshly cooked samples, and frozen samples reheated via microwaving exhibited a more intense flavour than pan-reheated or fresh mussels. This has important implications for consumer experience and the marketability of sous vide mussels. Compared with previous research, our study provides a new understanding of the sensory properties of sous vide mussels under different storage and reheating conditions.

### 3.5. The Effects of Food Additives on the Shelf Life of Sous Vide Cooked Mussels

The results of the optimisation of food products by means of food additives are shown in Figure 4, which illustrates the TVB-N and TBARS results of food additives for the prevention of spoilage and lipid oxidation in sous vide cooked mussels.

With the addition of preservatives, a significant decrease in TVB-N was observed on the sous vide cooked mussel samples during 28 days of storage. In Figure 4, the bar of the nitrogen content at 28 days is not shown as it exceeded the limits of acceptance (>35 mg of N/kg). However, for the samples with preservatives, none exceeded the limits of acceptability even after 28 days of storage. The samples with a combination of citric acid, potassium benzoate, and potassium sorbate (CM) reported the lowest amount of N for kg of fresh product after 28 days of refrigeration (22 mg of N/kg).

Concerning the TBARS experiments, our results showed that the use of all three preservative treatments led to a significant reduction in the rancidity of refrigerated sous vide mussels during 28 days of storage (*p* < 0.05). No significant differences were observed between samples during the first 7 days of storage. However, from day 14, increasingly marked differences were observed between the preserved samples and the control without additives. In fact, after 28 days of refrigerated storage, the MDA content of mussels in the control was almost 1.8 mg MDA/Kg, while the mussels stored with additives registered values 3 times lower (in the case of CM and BC). It is reported in the literature that citric acid is able to inhibit the rancidity development of Wels catfish fillets [49] and horse mackerel fillets during frozen storage [50]. Grape seed oil (GSO) was supplemented in the CM formulation. This oil also demonstrated high antioxidant properties and oxidative stability [51]. In fact, when adding GSO together with citric acid, a significant reduction in TBARS values was observed after 14, 21, and 28 d of storage (Figure 4B). Moreover, recently, studies have highlighted the significant antimicrobial activity of GSO. In particular, it has been found that the supplementation of GSO to chitosan film led to the inhibition of *Pseudomonas lactis* bacteria [52]. In our study, we confirmed several findings from previous research on the sous vide cooking of mussels, such as an improvement in microbiological quality with cooking temperatures of 80 °C or above. However, our study also revealed several unique insights. For instance, we found that the addition of specific food additives, namely citric acid, potassium benzoate, and potassium sorbate, in combination with grape seed oil, significantly reduced TVB-N and TBARS during 28 days of storage. This is significant because it indicates decreased spoilage and lipid oxidation, extending mussels’ shelf life. This finding adds nuance to previous research by demonstrating the potential of these specific additives in enhancing the quality of sous vide cooked mussels.

Based on the analysis results of the microbial load during the shelf life of the acidified products (Table 5), AB, BC, and CM showed a significant effect on the growth of mesophilic bacteria compared with the control without any food additives (Control).

This could also explain why the TVB-N value was lower in the treated samples than in the control, as the volatile nitrogen is mainly produced by spoilage bacteria. *Clostridium perfrigens, Bacillus cereus, Listeria monocytogenes*, and *Salmonella* spp. are among the bacteria with significant pathogenic risk in seafood [53], which were not detected in any of the treatments in this study. The inhibitory effect against bacterial growth using citric acid was also reported by other authors [40]. In fact, citric acid is a well-known natural antimicrobial compound, as the uncharged form of the acid can penetrate the cell membrane and acidify the cytoplasm of bacteria [54]. Regarding CM formulation, the addition of GSO increased the microbial stability of the mussels probably due to the antimicrobial properties of GSO [52].

## 4. Conclusions

This study offers insights into the influence of cooking temperature, freeze–thaw processes, and food preservatives on the physical properties of sous vide cooked mussels.

The key findings of this study include the dual-natured impact of cooking temperature on moisture content, with higher temperatures causing initial water loss but enhancing moisture preservation over time. Lower cooking temperatures (72 and 80 °C) were found to maintain colour stability, an important factor for product acceptance. In the context of freezing and thawing processes, our findings indicated that the employed methods could substantially impact the mussels’ physical properties. Specifically, fast freezing followed by slow thawing (FF-ST) yielded the highest moisture content, highlighting a potential strategy for better preserving the moisture in mussels and possibly improving their texture.

The investigation of food preservatives revealed their potential in extending the shelf life of sous vide cooked mussels. Citric acid was found to be the most effective preservative in reducing microbial growth, maintaining product quality, and extending shelf life compared with other preservatives. However, the preservative choice should also consider their potential effects on taste, texture, and overall consumer acceptability.

These findings underscore the need for further research on optimal cooking and freeze–thaw protocols for various seafood types, and a more detailed exploration of the synergistic effects of preservatives, cooking, and freeze–thaw processes. Overall, this study provides a scientific foundation for the seafood industry to develop best practices for the sous vide cooking and preservation of mussels.

## Figures and Tables

**Figure 1 foods-12-02900-f001:**
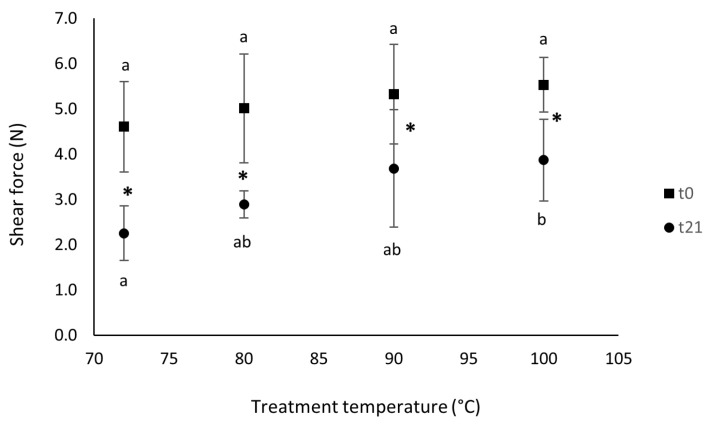
Shear force values of sous vide refrigerated (+4 °C) mussels cooked at different temperatures and measured at different times (■ t0; ● t21). Different letters indicate significant differences within groups (*p* < 0.05); asterisks indicate significant differences between groups.

**Figure 2 foods-12-02900-f002:**
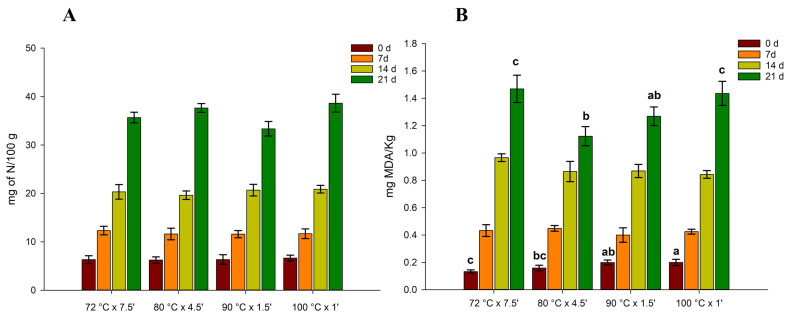
Changes in total volatile basic nitrogen (TVB-N expressed as mg of nitrogen/100 g of mussels, (**A**) and 2-thiobarbituric acid-reactive substances (mg of malondialdehyde/kg, (**B**) of four different thermal treatments of sous vide cooked mussels stored at 4 °C as a function of storage time. Different letters represents a significant difference (*p* < 0.05).

**Figure 3 foods-12-02900-f003:**
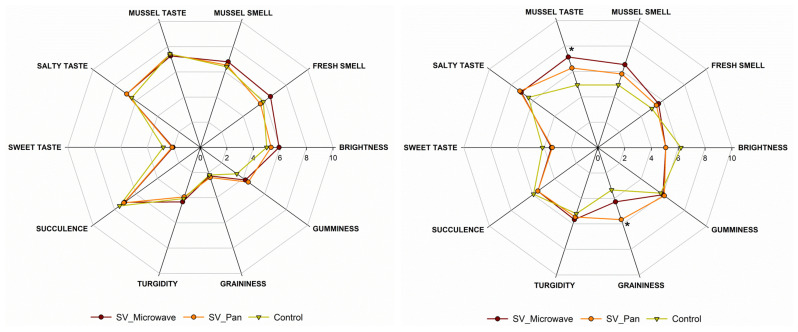
Spider plot of specific sensory attributes of refrigerated sous vide mussels (left) and frozen sous vide mussels (right). Control refers to fresh-cooked mussels. Asterisk (*) for each attribute indicates a significant difference (*p* < 0.05) compared with the control.

**Figure 4 foods-12-02900-f004:**
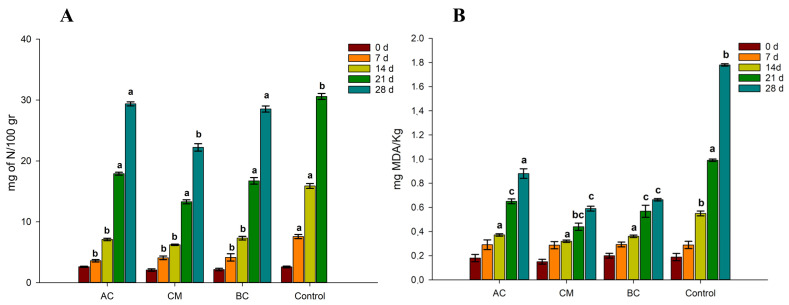
Effects of food additives on total volatile basic nitrogen (TVB-N, (**A**)) and 2-thiobarbituric acid (mg of malondialdehyde/kg, (**B**)) of mussels cooked sous vide during 28 days of refrigerated storage (+4 °C). AC = treatment using citric acid; BC = citric acid and grape seed oil; CM = treatment using citric acid, potassium benzoate, and potassium sorbate; Control = no food additives. Legends refers to the time of analysis (0 days to 28 days). Different letters represents a significant difference (*p* < 0.05).

**Table 1 foods-12-02900-t001:** Sensory attributes and references selected for the descriptive analysis of mussels.

Sensory Descriptors	References
**Appearance**	
Brightness	Weak: Opaque tile Strong: Shine tile
**Odour**	
Typical mussel odour	Weak: Absent Strong: Fresh raw mussel
Fresh odour	Weak: Tilapia Strong: Fresh raw mussel
**Texture**	
Turgidity	Weak: Overcooked mussel Strong: Cooked octopus
Gumminess	Weak: Overcooked mussel Strong: Cooked octopus
Succulence	Weak: Dry sponge Strong: Moistened sponge
Graininess	Weak: Well-washed mussel Strong: Mussel with sea-sand
**Taste**	
Salty taste	Weak: Salt-free mussel Strong: Salty mussel
Typical mussel taste	Weak: Absent Strong: Fresh mussel

**Table 2 foods-12-02900-t002:** Results of the colorimetric and moisture analysis of mussels cooked sous vide at different temperatures in PA/PP bags during the refrigerated storage at +4 °C (0 and 21 days).

Sample	Moisture %	L*	a*	b*
**T72**				
0 d	75.1 ^b^ ± 2.4	87.4 ^c^ ± 2.8	−1.6 ^a^ ± 0.6	18.4 ^c^ ± 1.4
21 d	70.2 ^a,^* ± 2.2	87.7 ^b,c^ ± 0.7	−2.0 ^a^ ± 0.3	20.6 ^c,b^ ± 0.6
**T80**				
0 d	72.4^a,b^ ± 1.2	91.3 ^b^ ± 0.5	−1.6 ^a^ ± 0.2	14.7 ^d^ ± 0.6
21 d	73.8 ^b^ ± 0.9	87.6 ^b,c^ ± 1.5	−1.5 ^a^ ± 0.4	21.4 ^b,^* ± 1.6
**T90**				
0 d	67.9 ^a^ ± 0.9	90.0 ^b,c^ ± 0.3	−2.2 ^a,b^ ± 0.2	15.6 ^d^ ± 0.5
21 d	74.4 ^b,^* ± 0.8	84.2 ^d,^* ± 0.6	0.4 ^c,^* ± 0.3	22.3 ^b,^* ± 0.8
**T100**				
0 d	70.1 ^a^ ± 3.3	93.6 ^a^ ± 0.1	−3.1 ^b^± 0.1	15.7 ^d^ ± 0.2
21 d	75.8 ^b,^* ± 0.7	76.9 ^e,^* ± 0.4	−0.9 ^a,^* ± 0.2	32.1 ^a,^* ± 0.9

Values are expressed as means ± SD (*n* = 5). T72= treatment at 72 °C; T80= treatment at 80 °C; T90 = treatment at 90 °C; T100 = treatment at 100 °C. Different letters indicate significant differences within groups in the same column; asterisks (*) indicate significant differences between groups (*p* < 0.05).

**Table 3 foods-12-02900-t003:** The results of the colorimetric and mechanical analyses of sous vide cooked (80 °C × 270 s) frozen mussels treated under different freezing conditions.

	L*	a*	b*	Moisture %	Shear Force (N)
**FF-FT**	68.4 ^c^ ± 0.4	−6.5 ^a^ ± 0.5	38.4 ^a,b^ ± 1.8	68.5 ± 1.7	5.0 ± 1.8
**FF-ST**	72.9 ^a,^* ± 0.9	−5.6 ^b,^* ± 0.3	41.4 ^a^ ± 1.2	71.9 ± 0.2	6.1 ± 3.2
**MF-FT**	71.8 ^a,b^ ± 0.4	−5.1 ^b,c^ ± 0.3	41.7 ^a^ ± 1.3	68.0 ± 1.8	4.0 ± 2.4
**MF-ST**	73.0 ^a^ ± 0.5	−4.6 ^c^ ± 0.1	38.4 ^a,b^ ± 0.9	70.2 ± 0.5	6.0± 3.9
**SF-FT**	68.9 ^b,c^ ± 0.9	−6.4 ^a^ ± 0.2	40.1 ^a,b^ ± 0.4	69.8 ± 2.3	4.7 ± 1.3
**SF-ST**	67.0 ^c^ ± 1.2	−6.3 ^a^ ± 0.3	37.5 ^b^ ± 1.1	67.4 ± 2.2	5.7 ± 0.9

Values are expressed as means ± SD (*n* = 5). FF-FT = fast freezing–fast thawing; FF-ST = fast freezing–slow thawing; MF-FT = medium freezing–fast thawing; MF-ST = medium freezing–fast thawing; SF-FT = slow freezing–fast thawing; SF-ST = slow freezing–slow thawing. Different letters indicate significant differences within groups in the same column; asterisks (*) indicate significant differences between groups (*p* < 0.05).

**Table 4 foods-12-02900-t004:** The effects of different sous vide cooking temperatures (72; 80; 90 and 100 °C) on the microbiological quality of sous vide cooked mussels during 21 days of refrigeration storage (Log CFU/g).

Total Mesophilic Bacteria	T0	7 d	14 d	21 d
**T72**	<1	1.21 ± 0.22	3.12 ± 1.43 ^a^	5.73 ± 1.12 ^a^
**T80**	<1	<1	1.08 ± 0.05 ^b^	2.49 ± 1.35 ^b^
**T90**	<1	<1	<1	2.16 ± 0.98 ^b^
**T100**	<1	<1	<1	1.78 ± 0.29 ^c^
**Total psychrophilic bacteria**	**T0**	**7 d**	**14 d**	**21 d**
**T72**	<1	<1	<1	2.95 ± 0.42 ^a^
**T80**	<1	<1	<1	2.15 ± 0.48 ^b^
**T90**	<1	<1	<1	1.97 ± 0.36 ^b^
**T100**	<1	<1	<1	1.94 ± 0.21 ^b^
** *Enterobacteriaceae* **	**T0**	**7 d**	**14 d**	**21 d**
**T72**	<1	<1	1.22 ± 0.32	3.2 ± 0.52 ^a^
**T80**	<1	<1	<1	1.25 ± 0.33 ^b^
**T90**	<1	<1	<1	1.64 ± 0.41 ^b^
**T100**	<1	<1	<1	1.44 ± 0.17 ^b^
***Pseudomonas* spp.**	**T0**	**7 d**	**14 d**	**21 d**
**T72**	<1	<1	<1	1.36 ± 0.11
**T80**	<1	<1	<1	<1
**T90**	<1	<1	<1	<1
**T100**	<1	<1	<1	<1
**Yeast count**	**T0**	**7 d**	**14 d**	**21 d**
**T72**	<1	<1	1.12 ± 0.27	2.43 ± 0.35 ^b^
**T80**	<1	<1	<1	1.18 ± 0.11 ^a^
**T90**	<1	<1	<1	1.25 ± 0.13 ^a^
**T100**	<1	<1	<1	1.02 ± 0.25 ^a^
**Mould count**	**T0**	**7 d**	**14 d**	**21 d**
**T72**	<1	<1	<1	2.53 ± 0.28 ^a^
**T80**	<1	<1	<1	1.21 ± 0.19 ^b^
**T90**	<1	<1	<1	1.33 ± 0.31 ^b^
**T100**	<1	<1	<1	<1
**LAB count**	**T0**	**7 d**	**14 d**	**21 d**
**T72**	<1	<1	<1	2.13 ± 0.45
**T80**	<1	<1	<1	<1
**T90**	<1	<1	<1	<1
**T100**	<1	<1	<1	<1

*Salmonella* sp. = absent in all the samples; *Listeria monocytogenes* = absent in all the samples; coagulase-positive staphylococci = absent in all the samples. Data are reported as averages of Log CFU/g ± standard deviation (*n* = 3); <1 means the result is under the detection limit. Different letters in the same column represent significant differences (*p* < 0.05).

**Table 5 foods-12-02900-t005:** The effects of food additives on the microbiological shelf life of cooked sous vide mussels during 28 days of storage at +4 °C (Log CFU/g).

Total Mesophilic Bacteria	T0	7 d	14 d	21 d	28 d
AC	<1	<1	1.08 ± 0.03 ^a^	2.15 ± 0.43 ^b^	3.91 ± 0.42 ^b^
BC	<1	<1	<1 ^b^	3.08 ± 0.32 ^a^	4.40 ± 0.33 ^a,b^
CM	<1	<1	<1 ^b^	2.05 ± 0.13 ^b^	3.35 ± 0.12 ^b^
Control	<1	<1	1.23 ± 0.13 ^a^	3.18 ± 0.32 ^a^	5.37 ± 0.35 ^a^

AC = treatment using citric acid; BC = citric acid and grape seed oil; CM = treatment using citric acid, potassium benzoate, and potassium ascorbate; Control = no food additives. Data are reported as averages of Log CFU/g ± standard deviation (*n* = 3). Different letters in the same column represent significant differences (*p* < 0.05). *Salmonella* sp. = absent in all the samples; *Listeria monocytogenes* = absent in all the samples; coagulase-positive Staphylococci = absent in all the samples.

## Data Availability

The data presented in this study are available on request from the corresponding author.

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
