# Peer review of "The Sous Vide Cooking of Mediterranean Mussel (*Mytilus galloprovincialis*): Safety and Quality Assessment"

_foods, 2023, doi:10.3390/foods12152900_

Round 1

Reviewer 1 Report

In general, I consider that the work offers a solution to a problem typical of the region, which is very important.

It must be remembered that in abstracts it is not correct to write abbreviations.

In the text, the first time they write an abbreviation they explain its meaning, and then continue writing both abbreviation and meaning. For example, on line 93: TVB-N. This should be corrected in all similar cases.

Did you consider that for the analysis of the organoleptic properties to be done with chefs? The above because the chefs have training on this.

It is very difficult to read and interpret the results by looking exclusively at the tables, perhaps the explanation given in the text should go before the tables. It is important to remember that a table must be able to be read and interpreted by itself, so the way in which the data is presented is essential. This can be substantially improved, including in the presentation by adding colors. 

For example, they write in their results less than one (with a symbol), but they do not explain what that means, it is not understood, do they mean that it is less than one UFC? that seems to mean.

Editing English is necesary. It must be revised, for example, in line 104 they write "in the laboratory" when, according to what they mean, they should write "at de laboratory"

Author Response

Dear referee,

We want to thank you for your time and suggestions to our manuscript.

The answers to your remarks are reported below:

-It must be remembered that in abstracts it is not correct to write abbreviations.

R: we improved the abstract, removing also the abbreviations.

- In the text, the first time they write an abbreviation they explain its meaning, and then continue writing both abbreviation and meaning. For example, on line 93: TVB-N. This should be corrected in all similar cases.

R: We fixed all the issues about the abbreviations

Did you consider that for the analysis of the organoleptic properties to be done with chefs? The above because the chefs have training on this.

R: We understand the reviewer’s doubt, but please consider the following information as the answer to your question. The sensory analysis we performed was a “descriptive” type. In this case, the evaluation (which is a quantification of the intensity of each selected attribute) must be performed by selected and trained assessors. That is the reason why before the real sessions, we conducted a selection of sensitive assessors followed by intense training, during which they were instructed about the meaning of the attributes, the evaluation techniques and the scales. We did not have already selected and trained chefs to involve in. Therefore, we selected 15 assessors from our usual panel and trained them according to the specific research study.

It is very difficult to read and interpret the results by looking exclusively at the tables, perhaps the explanation given in the text should go before the tables. It is important to remember that a table must be able to be read and interpreted by itself, so the way in which the data is presented is essential. This can be substantially improved, including in the presentation by adding colors. 

R: We thank you for your suggestions. We improved the table captions with more informations. For the Table 1 the text has been placed before the tables, but this is against the journal rules (which wants the table placed right after its mention in the text). However we tried to improve the readability of the tables.

For example, they write in their results less than one (with a symbol), but they do not explain what that means, it is not understood, do they mean that it is less than one UFC? that seems to mean.

R: for the microbiological analysis, when the plate were under the limit of detection, we reported <1 Log CFU/g, so it is not <1 CFU/g. We improved this in the text for clarity

Reviewer 2 Report

Please find in the following my comments about the review of a manuscript under the title (Sous-vide cooking of Mediterranean mussels (Mytilus gallo-provincialis): safety and quality assessment).

Originality and relevanc

§  The study is interesting for reading.

§  The study has moderate scientific quality.

§  The study is relevant to the scope of this journal.

§  The manuscript is relevant to the field and its presentation needs minor modifications to be clearer.

Comments:

 Abstract:

§  The abstract should be concise providing the main points in your manuscript. The provided one is deficient.

Introduction

§  The rationale of the study is not clear in the introduction section.

Results:

§  The title and legends of the figures should be informative and self-explanatory? Revise

Discussion:

§  For most of the discussion, you just confirmed that your results matched previously published data, please highlight the differences that your study brought out, or what new data it revealed.

§  Add a List of abbreviations.

Moderate editing of English language required

Author Response

Dear referee,

We want to thank you for your time and suggestions to our paper.

Here you will find all the detailed answers to your comments (answers in bold):

 Abstract:

  • The abstract should be concise providing the main points in your manuscript. The provided one is deficient.
  • R: We improved the abstract by adding other main findings of the study

Introduction

  • The rationale of the study is not clear in the introduction section.
  • R: Thank you for your feedback. We agree that the rationale for the study could be more clearly articulated in the introduction. In the revised manuscript, we have expanded the introduction to more clearly define the problem of mussel perishability and the need for improved processing methods. We have also discussed the limitations of current processing methods and stated our hypothesis that sous-vide cooking could improve the shelf life and quality of mussels. We have explained our experimental approach and discussed the potential impact of our research on mussel processing and quality. We believe these revisions provide a clear rationale for our study.

Results:

  • The title and legends of the figures should be informative and self-explanatory? Revise
  • R: we improved the table and figure captions for clarity

Discussion:

  • For most of the discussion, you just confirmed that your results matched previously published data, please highlight the differences that your study brought out, or what new data it revealed.
  • R: We have revised the discussion section of our manuscript to emphasize these points. We believe that these changes address your concerns and highlight the novel contributions of our study.
  • Add a List of abbreviations.
  • R: we followed the author guide of the journal, which do not requires a list of abbreviations. However if this is important and the editor asks for this we can certainly add it to the manuscript.

Reviewer 3 Report

The manuscript includes an interesting study. However, the way it has been presented is poor, in some sections even difficult to be understood.

Title

It would be better to employ the word “mussel”.

Abstract

Line 34: TBA levels ought to be replaced by TBARS levels or TBA value.

Line 36: Clarify units: mg of N/kg.

The title does not seem to agree with the Abstract section. The focus of the study, basic objective, ought to be clearly expressed. After reading the Material and methods section, it can be concluded that this section does not reflect the content of the study. It ought to be written again.

Keywords

bivalve and food technology could be eliminated. Contrary, other words ought to be including according to the real objective of the study.

Experimental part

Line 101: Replace design by procedure.

Line 154: Some justification for the quantities employed for the preservative compounds ought to be provided.

Line 151: TBA is not measured but the TBA-i or the TBARS content. This mistake is repeated in several parts of the manuscript.

Line 170: Some minor basic details on such methods ought to be provided

Line 201: Provide details on the 0-10 scores employed in the scale. What is the limit of acceptability ?

Results and discussion

Table 2: Colour parameters: L*, a*, and b* (also in Table 3).

Lines 262-274: Mention and discuss the relationship between b* and lipid oxidation development.

Line 346: Headings of sub-sections ought to be performed according to the scientific quality of the present journal. “TBA results” ??

Figure 2, A and B: Clarify units: /100 g and /kg of what ? mg of N ? or mg of TVB-N ? The same in Figure 4.

Line 374-394: Mention and/or correlate with b* colour values.

Conclusions

Some shortening of this section ought to be carried out.

MInor performances needed.

Author Response

Dear referee,

We want to thank you for all your time and for all the remarks and suggestions to our manuscript.

We will answer point by point (answer in bold) to your comments:

-It would be better to employ the word “mussel”.

R: we fixed this in the title

Abstract

-Line 34: TBA levels ought to be replaced by TBARS levels or TBA value.

R: We fixed this all along the manuscript

Line 36: Clarify units: mg of N/kg.

R: we fixed the units of TVB-N all along the manuscript.

The title does not seem to agree with the Abstract section. The focus of the study, basic objective, ought to be clearly expressed. After reading the Material and methods section, it can be concluded that this section does not reflect the content of the study. It ought to be written again.

R: Thank you for your valuable feedback. We agree that the title and abstract of our paper could be more clearly aligned with the focus of our study. However for what concerns the materials and methods we described all the experiments in 2.1 paragraph, highlighting  the experimental design in details.
For what concerns the chemical, physical, sensory and microbiological analysis we reported in details all the methods used with the respective reference.
If it is required we can fix any shortcoming from the material and methods if specified.
We fixed the abstract.

Keywords

bivalve and food technology could be eliminated. Contrary, other words ought to be including according to the real objective of the study.

R: Following your suggestion, we removed "food technology" and "bivalve" from the keywords and added these: "Shelf-life" and "seafood"

Experimental part

Line 101: Replace design by procedure.

R: we replaced design with procedure 

Line 154: Some justification for the quantities employed for the preservative compounds ought to be provided.

R: we processed the mussels using a quantity of preservatives lower than the legal limits imposed by the European community for each compound. We specified this in the text.

Line 151: TBA is not measured but the TBA-i or the TBARS content. This mistake is repeated in several parts of the manuscript.

R: we are sorry for the mistake, we fixed the term "TBA" with "TBARS" all along the manuscript

Line 170: Some minor basic details on such methods ought to be provided

R: we added some informations on the method used.

Line 201: Provide details on the 0-10 scores employed in the scale. What is the limit of acceptability ?

R: Many thanks for the suggestion, we modified lines 208-209 as: “Each characteristic was scored using a point scale (0–10), where 0 is “absent”, 10 is “very intense and 5 “moderate”. All the details for how to use the scale and the categories of the attributes have been provided in Table 1. Also, please consider that our sensory analysis was a “descriptive” type, so we just wanted to obtain a sensory profile/picture of the samples. In this case, we cannot talk about “acceptability” (the quantitative descriptive analysis does not include the acceptability evaluation) and there is no limit to take into account for the discussion. Each sample has been described from a sensory point of view, by obtaining a specific sensory profile. Data were discussed by comparing the intensity scores assigned to each evaluated attribute for all the samples.

Results and discussion

Table 2: Colour parameters: L*, a*, and b* (also in Table 3).

R: we are sorry for the mistake, we fixed these in all the tables.

Lines 262-274: Mention and discuss the relationship between b* and lipid oxidation development.

R: Thank you for the suggestion. In some studies, an increase in the b* value (indicating a shift towards yellow) has been associated with lipid oxidation. This is because lipid oxidation can lead to the formation of compounds that absorb light in the blue region of the spectrum, causing the food product to appear more yellow.
However, the relationship between b* value and lipid oxidation can be complex and may depend on various factors, such as the type of food product, the extent of lipid oxidation, and the presence of other pigments. 
We specified this in the text.

Line 346: Headings of sub-sections ought to be performed according to the scientific quality of the present journal. “TBA results” ??

R: We agree that some headings may not be clear or descriptive enough for the readers of the journal. We fixed some of them (specifically 3.2 and 3.3 headings)

Figure 2, A and B: Clarify units: /100 g and /kg of what ? mg of N ? or mg of TVB-N ? The same in Figure 4.

R: we apologize for the mistakes, TVB-N values are expressed as mg of N / 100 gr of mussels, we fixed this in the text and in figure 4 

Line 374-394: Mention and/or correlate with b* colour values.

R: thank you for the suggestion, we added this sentence to the manuscript:"This data was in line also with the colorimetric analysis (Table 2), were the highest b* value was observed with the mussels cooked at 100°C, denoting an increase of lipid oxidation".

Conclusions

Some shortening of this section ought to be carried out.

R: Thank you for your feedback on the conclusion section of our paper. We agree that this section could be more concise and focused. In the revised manuscript, we have shortened the conclusion section by focusing on the key findings of our study